# Fermentation Extract of Naringenin Increases the Expression of Estrogenic Receptor β and Modulates Genes Related to the p53 Signalling Pathway, miR-200c and miR-141 in Human Colon Cancer Cells Exposed to BPA

**DOI:** 10.3390/molecules27196588

**Published:** 2022-10-05

**Authors:** Sara Julietta Lozano-Herrera, Gabriel Luna-Bárcenas, Ramón Gerardo Guevara-González, Rocio Campos-Vega, Juan Carlos Solís-Sáinz, Ana Gabriela Hernández-Puga, Haydé Azeneth Vergara-Castañeda

**Affiliations:** 1Advanced Biomedical Research Center, School of Medicine, Universidad Autónoma de Querétaro, Querétaro 76140, Qro., Mexico; 2Cinvestav-Centro de Investigación y de Estudios Avanzados del Instituto Politécnico Nacional, Unidad Querétaro, Querétaro 76230, Qro., Mexico; 3Biosystems Engineering Group, School of Engineering, Autonomous University of Queretaro-Campus Amazcala, Highway Amazcala-Chichimequillas S/N, Km 1, Amazcala, El Marques, Querétaro 76265, Qro., Mexico; 4Research and Graduate Studies in Food Science, School of Chemistry, Universidad Autónoma de Querétaro, Querétaro 76010, Qro., Mexico

**Keywords:** colon cancer, ERβ, BPA, fermented extract of naringenin, miR-200c, miR-141

## Abstract

The estrogenic receptor beta (ERβ) protects against carcinogenesis by stimulating apoptosis. Bisphenol A (BPA) is related to promoting cancer, and naringenin has chemoprotective activities both can bind to ERβ. Naringenin in the colon is metabolized by the microbiota. Cancer involves genetic and epigenetic mechanisms, including miRNAs. The objective of the present study was to evaluate the co-exposure effect of colonic in vitro fermented extract of naringenin (FEN) and BPA, to elucidate molecular effects in HT-29 colon cancer cell line. For this, we quantified genes related to the p53 signaling pathway as well as ERβ, miR-200c, and miR-141. As an important result, naringenin (IC50 250 µM) and FEN (IC50 37%) promoted intrinsic pathways of apoptosis through phosphatase and tensin homolog (PTEN) (+2.70, +1.72-fold, respectively) and CASP9 (+3.99, +2.03-fold, respectively) expression. BPA decreased the expression of PTEN (−3.46-fold) gene regulated by miR-200. We suggest that once co-exposed, cells undergo a greater stress forcing them to mediate other extrinsic apoptosis mechanisms associated with death domain FASL. In turn, these findings are related to the increase of ERβ (5.3-fold with naringenin and 13.67-fold with FEN) gene expression, important in the inhibition of carcinogenic development.

## 1. Introduction

Colon cancer is one of the most common cancers worldwide, ranking third after breast cancer in women, prostate cancer in men, and lung cancer in both sexes [1,2]. This type of cancer is inversely related to dietary habits, especially the consumption of fruits and vegetables rich in fiber and antioxidant compounds [3]. Among antioxidants, flavonoids, especially flavonones such as naringenin, have become known for their antioxidant and anti-inflammatory effects against colon cancer through different mechanisms such as transactivation of estrogen receptors (ER) [4,5,6,7,8]. There are two different isotypes of ER, α (ERα) and β (ERβ), encoded by the ESR1 and ESR2 genes, respectively. The function of ERβ is related to apoptotic processes that reduce carcinogenic progress. ERα is associated with anti-apoptotic processes. Therefore, modulation of ERs activity and expression by estrogenic disruptors could have a preventive or promoting effect on colon cancer [9,10]. 

Some natural estrogenic disruptors, such as naringenin, can bind to ERβ, promoting its own expression and apoptotic processes, which has a protective effect on the colon. However, other disruptors of synthetic origin, such as BPA, a compound used to polymerize plastics used in various products such as food containers, can migrate into the food matrix and be ingested by the consumer, promoting colon cancer because they have an antagonistic effect on ERβ in the presence of estradiol [11,12]. Interestingly, simultaneous exposure to both types of disruptors, naringenin and BPA, has been studied in the development of breast cancer, and naringenin had an antagonistic effect on ERα associated with proliferation processes even in the presence of simultaneous exposure to BPA [13,14,15]. 

The antiproliferative or apoptotic molecular mechanisms of ER are diverse, including signalling associated with the p53 pathway; there are also epigenetic mechanisms such as miRNAs that can interfere with the expression of genes. The miRNAs are small RNA fragments with a length of 18–25 nucleotides that primarily inhibit the translation process by binding to the 3′UTR region of mRNA. Their activity is associated with the promotion of carcinogenesis once they inhibit the translation of tumour suppressor genes. In this regard, miR-200c has been reported to play a key role in suppressing tumour progression by inhibiting epithelial–mesenchymal transition and metastasis in various cancer [16]. 

Naringenin is a flavonoid that belongs to the subclass of flavanones. It is found in citrus fruits such as oranges, grapefruits, and tomatoes. After digestion, a large amount of naringenin enters the colon, where it is metabolised by the resident microbiota through a fermentation process. This process produces by-products such as hippuric acid, 3(3-hydroxyphenyl) propionic acid, and 3(3,4-dihydroxyphenyl) propionic acid, or phloroglucinol, which have been reported previously [17,18,19]. These compounds have antioxidant and anti-inflammatory activity [20,21]. It is therefore hypothesized that consumption of foods rich in flavonoids such as naringenin could attenuate the negative effects of BPA, but information on their effects as estrogenic disruptors is lacking. Therefore, the aim of this study was to investigate the anticarcinogenic effect of naringenin and its fermented extract from colon upon simultaneous exposure to the disruptor BPA in human adenocarcinoma cells HT-29, and to elucidate the molecular mechanisms involved. 

## 2. Results

### 2.1. Products Generated by Microbial Metabolic Fermentation from Naringenin

FEN was analyzed by the method UPLC-MS, identifying naringenin and other compounds such as 3(3-hydroxyphenyl) propionic acid (3-HPPA), apigenin, phenylacetic acid, and secoisolariciresinol (Table 1). Nevertheless, only apigenin and 3-HPPA can be considered as by-products of naringenin, since the other two compounds were also detected in the blank, the culture medium used for fermentation. Several authors have found dehydroxylations, hydroxylations, and deglycosylations of phenolic compounds generated by microbial metabolism to produce various byproducts, as we show (Figure 1) [22,23]. 

#### Antioxidant Capacity of FEN and Naringenin

The antioxidant capacity of both naringenin and FEN is shown in Table 1. The results show a decrease in the antioxidant capacity of FEN by DPPH, a synthetic radical that is not similar to the conditions prevailing in the organism; therefore, the ORAC method is the most accurate to obtain information at the physiological level. 

Regarding the ORAC method, there is no statistical difference between the antioxidant capacity of FEN and naringenin, so they have similar biological capacity to transfer hydrogen atoms. 

### 2.2. Naringenin and Its Fermented Extract Decrease Cell Viability in Co-Exposition with BPA

Cell viability experiments were performed to determine the effects of naringenin and FEN on HT-29 cells. Cells treated with naringenin showed a decrease in cell viability with a dose–response effect. Naringenin successfully suppressed cell growth, with an IC50 value of 250 µM at 24 h of exposure time (Figure 2a). To determine the IC50 of FEN (37%), cell viability was measured at concentrations of 25–43% of FEN (Figure 2b) after 24 h and a linear decrease in cell viability was also observed. Cells treated with BPA showed no significant effect on cell viability under our study conditions, but the responses at the molecular level were different, but cell viability decreased as much with simultaneous exposure to BPA (4.4 µM) + naringenin or +FEN, as it did without BPA (Figure 3).

#### 2.2.1. Naringenin and FEN Induce Apoptotic Cell Death and BPA Induces Necrosis in HT-29 Cells

Flow cytometry assays with Annexin V were performed to evaluate cell death. Naringenin (250 µM) induces death in 52 ± 0.7% of cells, of which 41 ± 0.7% correspond to apoptosis, with or without co-exposition to BPA (4.4 µM), and 7 ± 1.3% of cells die by necrosis (Figure 4). Cells treated with FEN maintain the trend of viability without (54 ± 0.3%) or with BPA (59 ± 0.7%), dying mainly by apoptosis (39 ± 1.5 and 37 ± 1.5%; respectively) (Figure 4).

In addition, when the cells were treated with BPA alone, 92% of cells died by apoptosis and the other 7% died by necrosis, a type of cell death that causes cell damage and inflammatory processes. To confirm cell death by necrosis, LDH assay, an indirect biomarker for necrosis, was performed (Figure 5). Necrosis was detected in 7–12% of cells under the different treatments, which is consistent with our results from the flow cytometer. 

#### 2.2.2. Naringenin and FEN Treatments Increase SOD Activity and Decrease the Amount of GSH

To determine the modulation of the endogenous antioxidant system by the experimental treatments, the activity of SOD enzyme and the amount of GSH were measured in the previously described treatments. Cells treated with FEN showed the highest percentage of O_2_ inhibition, consistent with the high activity of SOD (13.18 ± 2.0%) (Table 2). 

On the other hand, GSH, the most potent antioxidant in the body, was significantly decreased by naringenin and naringenin + BPA (110.75 ± 1.5 and 129.19 ± 0.9 Nmol/mL, respectively) (Table 2), suggesting that GSH, as an electron donor, undergoes oxidation–reduction reactions to neutralize free radicals generated by oxidative stress in the cell.

#### 2.2.3. Naringenin and FEN Increase the Expression of ERβ Activating Apoptotic Genes, whereas BPA Increases ERα Expression, Related to Survival

The relative mRNA expression of ERβ and GPR30 in HT-29 cells is shown in Figure 6. Our results show that naringenin and FEN treatments increase the ERβ expression by 5.3 ± 0.77-fold and 13.67 ± 2-fold, respectively, while BPA-treated cells increase by 2.53 ± 0.2 fold compared with the negative control. 

On the other hand, GPR30 is an estrogen-sensitive G protein-coupled receptor that can trigger various signalling pathways such as proliferation, apoptosis, and cell migration. In the current study, negative transcriptional regulation of GPR30 by naringenin and BPA was detected (Figure 6b). The decrease in GPR30 receptor expression needs further investigation to identify the triggering mechanism.

The expression of genes related to the p53 pathway was determined using a real-time PCR array, and the results are presented in the form of fold-change of expression levels (Table 3). As shown, some important genes involved in the apoptosis process are up- or down-regulated. This is the case with the overexpression of the Caspase 9 gene induced by the treatments with naringenin (3.99-fold), FEN (2.03-fold), naringenin + BPA (2.36-fold), and FEN + BPA (12.27-fold).

Bcl-2 is an antiapoptotic gene whose expression was decreased by naringenin (-2.17-fold) but not by FEN (1.70-fold). It was increased by naringenin + BPA (40.54-fold) and FEN + BPA (146.89-fold) treatments, suggesting activation of BPA in co-exposure responsive mechanisms to prevent apoptosis, which are triggered only in the presence of naringenin or its metabolites. 

Activation of both extrinsic and intrinsic apoptosis pathways is suggested by the co-exposure treatments in the present study, as an increase in FasLG gene expression with naringenin + BPA (35.45-fold), FEN (4.63-fold), and FEN + BPA (52.37-fold), the Fas gene in cells treated with naringenin + BPA (12.09-fold), FEN + BPA (15.3-fold), and FADD (1.19-fold) and (2.05-fold), respectively. 

Another gene involved in apoptosis is caspase-2 and CRADD [24]. Naringenin increased the expression of caspase-2 (2.14-fold) and CRADD (2.96-fold). In addition, TNFRSF10D was also overexpressed by naringenin (4.98-fold), naringenin + BPA (14.79-fold) and FEN + BPA (9.7-fold). 

PTEN is involved in the modulation of several cancer processes, including apoptosis [25,26]. In the current study, a reduction in PTEN was observed with BPA treatment (-3.46-fold), but co-exposure with naringenin (1.20-fold) or FEN (6.98-fold).

Co-exposure treatments naringenin + BPA (21.69-fold), FEN + BPA (36.22-fold), and FEN (2.85-fold), increased MDM2 gene expression. In addition, overexpression of the TP73 gene was observed among treatments with naringenin (3.99-fold), naringenin + BPA (75.89-fold), FEN (2.65-fold), and FEN + BPA (101.66-fold). 

Once activated, TP73 can bind to PTEN, which was overexpressed under naringenin treatment (2.70-fold), FEN (1.72-fold) and down-regulated under BPA treatment (-3.46-fold); this binding activates BBC3 (for Bcl-2 binding component 3 or PUMA), which is negatively regulated by BPA (−4.96-fold). 

Our results show an up-regulation of MLH-1 by naringenin treatment (4.48-fold) and a down-regulation by BPA (−7.94-fold). 

In this study, we also found a decrease in ATR expression in HT-29 cells treated with BPA (-2.16-fold). Therefore, this might not have a beneficial effect on cell cycle arrest; whereas co-exposed to naringenin (3.28-fold) or FEN (5.63-fold) resulted in an increase in expression, suggesting a contribution to cell cycle arrest, as previously informed for apigenin, which is present in FEN. 

In addition, RPRM is overexpressed in BPA-treated cells in the current study (3.63-fold), and even more strongly in naringenin + BPA (63.17-fold), FEN (5.11-fold), and FEN + BPA (33.01-fold) treatments. 

MYOD is a gene involved in this process. Our results showed that cells treated with naringenin (3.99-fold), naringenin + BPA (75.89-fold), FEN (5.20-fold), and FEN + BPA (26.82-fold), had an increase in MYOD expression, while conversely BPA treatment (-1.10-fold) decreased MYOD expression (−1.10-fold). 

The results support intrinsic and extrinsic apoptosis in cells treated with naringenin and FEN and suggest the relationship with ER. On the other hand, they support the malignant mechanisms of BPA treatment.

#### 2.2.4. miR-200c Regulates PTEN Protein Expression

Regarding to miRNAs expression, both miR-200c and miR-141, which belong to the miR-200 cluster were analyzed (Figure 7). An increase in the expression of miR-200c was observed under BPA treatment (2.94 ± 0.44-fold) and a lower expression of miR-141 was observed under FEN treatment (0.01 ± 0.01 fold). 

The RT^2^ Profiler PCR Data Analysis software from GeneGlobe, Qiagen, suggests PTEN as a target gene, and we ratify with miRmap software. This software presents the results in an algorithm that is thermodynamic, evolutionary, probabilistic, and sequence-based, finally giving a score from 0–100. Our miRNAs have 63.52 (miR-200c) and 90 (miR-141) with PTEN gen0065 (Figure 8). 

## 3. Discussion

The results obtained in this study show the generation of 3-HPPA by c-ring cleavage and of apigenin by dehydrogenation of naringenin during the fermentation process. The formation of these compounds and others such as apiferol, eriodyetol, or some hydroxycinnamic acids by hydroxylations, hydrogenations, hydrolysis, or methylations depends on the composition of the microbiome in the colon. The above findings are relevant because these compounds have antioxidant capacities and activities associated with disease prevention [22,23,27].

FEN exhibited a lower antioxidant capacity than naringenin as measured by DPPH. This is probably because the by-products found in FEN differ from naringenin in the amount and position of the hydroxyl groups. Apigenin, a byproduct of FEN, has a higher antioxidant potential than naringenin because of the 2,3-dehydrogenation reaction leading to conjugative C=C binding. However, the interaction of different metabolites leads to antagonism that reduces the ability to donate electrons to the DPPH radical, which is the fundamental basis of the assay. Second, in the evaluation of FEN and naringenin by ORAC method, which is based on the ability of the molecule to transfer hydrogen atoms to oxygen radicals, there is no difference between FEN and naringenin [28,29]. Thus, it can be seen that naringenin has a great ability to transfer electrons and has the same ability to transfer hydrogen atoms to oxygen radicals as the group of by-products formed after fermentation. 

The antioxidant capacity of naringenin shows how it can interact with some proteins, either by reduction ROS, chelation of metals, or in a pro-oxidant manner, which in a cell metabolism altered by cancer would promote apoptosis and eventually inhibition of cell proliferation. In addition to these mechanisms, naringenin has shown several others by which it inhibits proliferation. These include some signaling pathways such as EGFR/MAPK, Akt, and the Wnt/β-catenin pathway, among others [30,31]. 

Naringenin (IC_50_ between 180–360 µM) has been shown to have an inhibitory effect on the proliferation of HT-29 colon cancer cells [32].

Currently, there is no evidence that FEN has the same effect. This is the first study to investigate the antioxidant capacity of the fermentation extract of naringenin in colon cancer cells. After gastrointestinal digestion, about 84% of naringenin enters the colon bound to an indigestible carbohydrate or in free form. Moreover, naringenin is one of the by-products of quercetin, another widely used flavonoid, so its transformation during the fermentative process is very important to determine an effect closer to the biological [33,34]. Some authors have reported the formation of diverse by-products such as 3-HPPA, which has been shown to have antioxidant and anti-inflammatory effects and also to promote apoptotic processes [34].

Within the antioxidant capacity, the activity of enzyme systems is very important, one of them is SOD, whose biological function is to dismute the peroxide anion into hydrogen peroxide. There are three SOD isoenzymes, Cu/Zn SOD (SOD1), which is found in greater quantity in the cytoplasm, Mn SOD (SOD2), located in the mitochondria, and EcSOD (SOD3), which is found mainly in the extracellular space. SOD3 is associated with inhibition of tumour growth and metastasis, while SOD1 and SOD2 are elevated in different stages of several cancers, including colon cancer, and can trigger proliferative and apoptotic signals depending on the amount of H_2_O_2_ produced [35,36]. In our assay, the total activity of SOD was measured, so we hypothesize that the increase in activity leads to H_2_O_2_ formation that promotes apoptosis under the FEN treatment.

Another antioxidant system is GSH. Although it is one of the most potent antioxidant systems, large amounts of GSH have been associated with metastasis and chemoresistance in cancer, as an adaptive response of the cancer cell to large amounts of free radicals. We hypothesize that GSH depletion after naringenin treatment, as reported by other authors, is mediated by multidrug resistance protein 1 (MRP1), which promotes increased sensitivity of cancer cells [6,30,37,38].

In the cells treated with BPA, it was observed that the viability of the cells did not decrease significantly and the cells that diet were mostly due to necrosis. Previously, it was reported that testicular cancer cells treated with BPA (0.01–10 µM µM) and exposed for 24 h maintained the same trend of decreasing cell proliferation (>25%); however, at concentrations of 10 µM an increase in cell migration (wound assay) was detected, in addition to increased expression of 50 migration-related proteins such as Gal-1, supporting the results of the current study, in which molecular responses of migration were observed under BPA treatment [39].

On the other hand, naringenin and FEN promoted apoptosis related to signalling pathway. Other reports suggest that naringenin (10 µM) induces apoptosis via the p38/MAPK pathway, which is modulated by ERβ agonist activation in DLD-1 colon cancer cells [40,41]. Moreover, simultaneous exposure of naringenin (100 µM) and BPA (10 µM) in breast cancer cells has shown an apoptotic effect due to a decrease in anti-apoptotic Bcl-2 proteins, mechanisms that may also play a role in the present study [42]. Moreover, naringenin and the by-products of FEN could promote the formation of ROS, which is related to the induction of the apoptotic process [43,44].

The differential expression of ERβ under naringenin and FEN treatment is the result of the position at which the ligands bind to the receptors and determine their positive or negative modulation. Kuiper et al. described in terms of relative affinity units (RBA) with respect to 17β-estradiol, that BPA had similar affinity for both estrogen receptor α and β, naringenin had higher affinity for ERβ and similarly apigenin, which is a component of FEN, so it can be assumed that the affinity of flavonoids for this receptor positively modulates its expression [45]. The reduction of ERβ expression has been associated with poor prognosis in cancer, since its main function is to promote the apoptotic process. [46,47] The lack of regulation of ERβ mRNA by BPA treatment is consistent with the reports of Hess–Wilson et al. on 1nM BPA-treated LNCaP prostate cancer cells [48]. 

In our study, treatment had no effect on GPR30. Dong et al. showed that BPA (10 µM) in breast cancer cells activates the Erk1/2 signalling pathway and transcriptional regulation of c-fos through GPR30 [49]. In addition, the flavonoid compounds genistein and baicalein were reported to activate GPR30, suggesting an antiproliferative effect. [50,51] Under the conditions studied, this receptor is not related to the results obtained.

We found high expression of caspase 9 in naringenin and FEN, suggesting that mitochondrial apoptosis may be activated by the stimulation of ROS, which may be related to the increase seen in SOD activity by FEN and the low amount of GSH by naringenin. In addition, naringenin decreases BCL-2 expression. Kang et al. demonstrated the fisetin, a flavone with antiangiogenic and antioxidant activity, increases caspase-9 activity and decreased Bcl-2 protein expression in NCI-H460 lung cancer cells at 75 μM for 24 h [33].

Our results show extrinsic apoptosis when BPA is co-exposed to naringenin or FEN. Lee et al. observed that kaempferol (40 and 60 μM, 48 h, HT-29 colon cancer cell line) promotes apoptosis through both extrinsic and intrinsic pathways. This process occurs through an increase in FAS-L protein expression, which activates caspase-8 and eventually leads to cleavage of Bid and activation of caspase-3, and through the increase in released cytochrome C, which activates caspase-9 and caspase-3 [52].

FASLG is a gene that is not constitutively expressed in the HT-29 cell line, and tumour necrosis factor alpha (TNF-α) has been shown to induce the expression of functional FasL on the cell surface of HT29 cells, so it may be able to induce apoptosis through this pathway. [53] This is related to the results shown, in which an increase in TNF expression is observed under treatments with naringenin + BPA, FEN, and FEN + BPA.

Other studies reported that polyphenols such as crocetin (100 μM) promote the FAS/FADD interaction in HT29 cells in a p73-dependent manner. This interaction activates BID, which subsequently activates apoptosis via mitochondrial [54].

Another gene involved in apoptosis is caspase-2, which encodes a protein involved in apoptotic and inflammatory mechanisms. CRADD is a gene associated with the pathway triggered by caspase-2. It encodes the protein containing the caspase recruitment domain and death domain (DD) (also known as RAIDD), which together with PIDD1 are necessary for the activation of caspase-2 as a pro-apoptotic protein. [24,54,55,56,57] In addition, naringenin increases the expression of caspase-2 and CRADD. Moreover, TNFRSF10D was also overexpressed by naringenin, naringenin + BPA, and FEN + BPA. TNFRSF10D is a gene encoding TRAILR1 receptor, which is associated with the promotion of proliferation processes as apoptosis and could regulate the extrinsic pathway of apoptosis [24,53].

Another important gene for promoting apoptosis is p53. This gen is post-translational activated by cellular stress regulated by MDM2 in the signalling cascade of mitochondrial apoptosis, which increases transcription of MDM2, resulting in inhibition of p53 activity. However, the p53 gene is mutated in the HT-29 cell line, causing the cell to activate other signalling pathways to reduce its viability [58]. Zeng et al. showed that MDM2 can modulate the activation of TP73 but not its expression in cells with mutated p53 gene [59]. The co-exposure treatments naringenin + BPA, FEN + BPA, and FEN increased MDM2 gene expression. In addition, overexpression of TP73 gene was observed among naringenin, naringenin + BPA, FEN, and FEN + BPA. Dabiri et al. investigated the role of TP73 in HT-29 cells and found the presence of TP73 in both the nucleus and cytosol of cells treated with bortezomil, a chemotherapeutic agent used as a proteasome inhibitor and promoter of cellular cycling and apoptotic death. The authors suggested that the role of TP73 in cellular apoptosis was due to the presence of a mutation in p53 upon treatment with naringenin and FEN [60]. 

Once TP73 is activated, it can bind to PTEN, which was overexpressed under naringenin treatment, FEN and down-regulated under BPA treatment; this binding activates BBC3 (for Bcl-2 binding component 3 or PUMA), which is negatively regulated by BPA. BBC3 is a proapoptotic gene encoding the BBC3 protein located in mitochondria and released into the cytosol by mitochondrial permeabilization along with other proapopototic molecules and cytochrome C, dependent or independent of p53 [61]. In HT-29 cells harbouring a p53 mutation, its activation by Sp1 and p73 was detected [62]. Tili et al. found overexpression of PTEN protein in SW480 cells treated with resveratrol (50 μM), which triggered apoptosis processes, similar to those observed when treated with naringenin and FEN [63].

On the other hand, Li et al. reported a decrease in PTEN protein expression in breast cancer cells treated with 10 μM BPA, related to an increase in cell proliferation and a higher percentage of cells in the S phase of the cell cycle, while 1 μM curcumin counteracted this effect [8]. In the current study, a reduction in PTEN was observed with BPA treatment, but the co-exposure with naringenin, or FEN, offsets this effect. PTEN is also involved in the modulation of cell adhesion, migration, and invasion through inhibition of the adaptor protein Shc and the focal adhesion protein kinase Fak. [64,65,66,67,68] Overexpression of the PTEN gene by treatments with naringenin and FEN suggests a decrease in the metastatic phenotype of tumour cells through inhibition of cell migration.

We hypothesise that the low expression of PTEN in BPA treatments is due to epigenetic modulation, i.e., high expression of miR-200c or miR-141. 

Chen et al. demonstrated the binding of miR-200 homologs to the 3’UTR region of PTEN in endometrial cancer cells using a luciferase assay, and the importance of E2 in regulating miR-200c expression was also demonstrated via ERα overexpression, which is important for the development of this type of cancer [65]. Under our experimental conditions, BPA induced miR-200c expression and PTEN gene repression, so it could increase cellular proliferation through a mechanism modulated at the molecular level by increasing ERα expression (ESR1). Mei et al. demonstrated the immunosuppressive role of miR-200c through the inhibition of PTEN and the increase of MDSCs (ROS in myeloid-derived suppressor cells), which suppress the antitumor immune response, suggesting various negative effects of BPA through the suppression of PTEN mediated by miR-200c, such as the inhibition of apoptosis, the increase of cell cycle and immune responses [66]. 

PTEN is also a target gene of miR-141, and studies performed on cells from patients with nasopharyngeal carcinoma have shown that miR-141 expression decreases PTEN expression of after treatment with cisplatin, related to chemoresistance [67].

The MLH-1 gene is one of the genes belonging to MMR, which encodes proteins that recognize and repair errors in microsatellite sequences of newly synthesized DNA. Therefore, its downregulation by methylation or mutation is common in colon cancer [69]. Our results show up-regulation of MLH-1 up-regulation by naringenin treatment and down-regulation by BPA. Lu et al. observed an increase in MLH-1 gene in HT-29 and SW480 cells treated with 17β-estradiol, suggesting a role of ERβ as an upstream mechanism [70]. We suggest positive regulation of MLH-1 by ERβ under naringenin treatment.

DNA damage is also detected by the ATM-Rad3-Related gene (ATR), whose activation is arrested at G1, S, or G2 stages of the cell cycle, a process mediated by the action of the checkpoint kinases CHK1 and CHK2 [71]. In this study, we also found a decrease in ATR expression in HT-29 cells treated with BPA. Therefore, this might not have a positive effect on cell cycle arrest, while the co-exposure to naringenin or FEN resulted in an increase in expression, suggesting a contribution to cell cycle arrest as previously informed for apigenin, which is present in FEN. 

On the other hand, Reprimo (RPRM) is a gene encoding a protein involved in cell cycle arrest in G2/M phase by regulating the activity of Cdc2 and cyclin B1 in a p53 and p73- dependent manner, and its loss of expression has been associated with more invasive stages of gastric cancer [72,73]. In the present study, all treatments except BPA, increased the expression of RPRM, which mediates cell cycle arrest. In addition, hypermethylation of RPRM related to ERα (ESR1) has been observed in several cancers such as breast cancer [74,75,76,77], which is overexpressed in cells treated with BPA in the present study, and even more so in treatments with naringenin + BPA, FEN, and FEN + BPA.

ERα expression, implicated in the development and progression of colon cancer, has been associated with lower survival in epidemiological studies [78]. Huang et al. have evaluated different concentrations of BPA (0.1–1000 nM) in prostate epithelial cells and reported 98% viability but with an increase in the expression of both α and β estrogenic receptors [79]. In the present study, the increase in ERα gene was found to be under BPA treatment, which is explained by the affinity that naringenin has for the ERα as the apinenin contained in FEN. Ye et al. support in their *in silico* analysis that the number of hydroxyl groups and their position affect the affinity for the receptor, probably the other naringenin byproducts present in FEN have synergy with the receptor [80]. It is important to highlight that ERβ expression is predominant in differentiated normal colon epithelium, but its expression decreases due to the hypoxic microenvironment in the colon when malignancy resulting from cancer progresses [81]. Therefore, naringenin and FEN treatments induced ERβ expression (Figure 6), which could promote apoptotic processes (Figure 4), even if there is an increase in ERα expression. 

SIRT1 is a nicotinamide adenosine dinucleotide (NAD/NADH) dependent histone deacetylase (HDAC) that acts by deacetylation of histones (H1, H3, and H4) and is associated with gene expression of ER [80,81]. Transcription of SIRT1 is regulated by FOXO3A, p53, E2F1 and other transcription factors [80]. This explains the increase in E2F1 and the increase in SIRT1 in co-exposure treatments. In ovarian cancer cells, loss of SIRT1 mRNA expression is associated with higher expression of ERβ [82]. Therefore, SIRT1 could possibly act as a coactivator or repressor of the different ERs depending on the cell line and its conditions, which could support the low expression of SIRT1 gene with the low expression of ERβ and high expression of ERα genes under the treatment with BPA. However, this mechanism does not apply to the other treatments; therefore, the role of SIRT1 is not fully elucidated [82,83,84]. 

In addition to the processes of apoptosis and cell cycle inhibition, the inhibition of metastasis is an important mechanism mediated by flavonoids. MYOD is a gene involved in this process by decreasing mRNA expression of E-cadherin and vimentin, both promoters of epithelial-mesenchymal transition (EMT) [85]. Our results showed that cells treated with naringenin, naringenin + BPA, FEN and FEN + BPA exhibited an increase in MYOD expression, and conversely, BPA treatment decreased MYOD expression. The promotion of metastasis mediated by BPA and its inhibition by naringenin and FEN in HT-29 cells is thus demonstrated in addition to the PTEN gene results.

## 4. Materials and Methods

### 4.1. Materials

Naringenin (#N5893), BPA (#239658), 3-(4,5-dimethylthiazol-2-yl)-2,5-diphenyltetrazolium bromide (#M2128), casein peptone, HCl, NaCl, KCl, MgSO_4_, KH_2_PO_4_, NaHCO_3_, CaCl_2_, NaOH, pepsin, pancreatin, Tween salts, hematin, glucose, crystal violet, formaldehyde, ethanol, LDH assay kits (#MAK066), SOD assay kits (#19–160), were purchased from Sigma-Aldrich Co. (St. Louis, MO, USA). HT-29 cells were purchased from the American Type Culture Collection (ATCC). Dulbecco’s modified Eagle’s medium (DMEM), fetal bovine serum (FBS), antibiotic-antimycotic, trypsin, and ethylenediaminetetraacetic acid (EDTA) were purchased from Gibco or Thermo Fisher Scientific, Inc. (Waltham, MA, USA). Anexin-V kit (#602128), were purchased from Millipore, Merk. Silica-gel columns kit (#PP-210S) and SCRIPT cDNA kit (#77176) were purchased from Jena Bioscience GmbH (Jena, Germany). Radiant TM Green Hi-ROX qPCR kit (#QS2005) were purchased from Radiant molecules, Alkali Scientific Inc. Finally, the Human RT2 Profiler Real Time PCR Array System (PAHS-027A) was purchased from Qiagen, SABiosciences, USA.

### 4.2. In Vitro Gastrointestinal Digestion and Fermentation of Naringenin

In vitro simulation of the digestion process of naringenin in the mouth, stomach, and small intestine was performed according to the procedure of Campos-Vega et al [86]. Briefly, 5 mL of saliva was collected from four volunteers, two males and two females, healthy, with no chronic degenerative diseases or contact infections in the past 30 days, no use of antibiotics, medications in general, or alcohol in the past 20 days. Subjects had to be of legal age and not eating a vegetarian or keto diet. The sample was diluted with 10 mL of distilled water. The samples were incubated with 1 g naringenin for 5 min. Samples were then adjusted to a pH of 2.0 with a 150 mM HCl solution to simulate gastrointestinal tract conditions. Pepsin (0.055 g) was dissolved in 0.94 mL of 20 mM HCl solution, and the samples were incubated at 37 °C for 2 h. Later, intestinal conditions were simulated with 3 mg of bovine bile and 2.6 mg pancreatin mixed with 5 mL of Krebs-Ringer buffer (118 mM NaCl, 4.7 mM KCl, 1.2 mM, MgSO_4_, 1.2 mM KH_2_PO_4_, 25 mM NaHCO_3_, 11 mM glucose and 2.5 mM CaCl_2_; pH 6.8). This solution was gasified with a gas mixture (10:10:80, H_2_:CO_2_:N_2_). 

The fermentation process in the colon was simulated according to the method of Campos-Vega et al [87]. Briefly, human stool inoculum from healthy donors (two males and two females) was used. Three grams of the stool sample was homogenised with 27 mL of phosphate buffer pH 7.0. One millilitre of the stool solution was collected and added to sterile basal culture medium pH 7.0 containing: 2 g/L peptone water, 2 g/L yeast extract, 0.1 g/L NaCl, 0.04 g/L K_2_HPO_4_, 0.04 g/L KH_2_PO_4_, 0.01 g/L MgSO_4_-7H2O, 0.01 g/L CaCl_2_-2H_2_O, 0.01 g/L NaHC32, 0.5 g/L L-cysteine, 0.5 g/L bile salts, 2 mL/L Tween-80, and 5 mL hematin solution (0.2 g dissolved in 5 mL NaOH). Finally, 1 mL of the sample obtained from in vitro gastrointestinal digestion was added, and the samples were placed in a fermenter containing a gas mixture (10:10:80, H_2_:CO_2_:N_2_) and allowed to ferment for 24 h at 37 °C, using raffinose as a fermentation control (100 mg, R0514, Sigma-Aldrich, St. Louis, MO, USA) [87]. The obtained extract was designated as fermentation extract of naringenin (FEN) and the basal nutrient medium as blank. 

#### 4.2.1. Identification of Metabolites

Fermentation-derived metabolites were identified according to the modified methodology proposed by Abu-Reidah et al [88]., using ultrahigh performance liquid chromatography (UPLC) (Waters UPLC Acquity I-class with PDA detector) coupled to a time-of-flight mass spectrometer VION QtoF. Leucine enkephalin was injected at 200 pg/μL at a flow rate of 10 μL/min. Injections were performed in MSe mode. Data were analysed using the UNIFI 1.9 program with Small Molecule Option.

#### 4.2.2. Antioxidant Capacity

The trolox equivalent antioxidant capacity (TEAC) of FEN, blank, of FEN and naringenin was evaluated by ORAC and DPPH assay. For the DPPH assay, the sample was incubated in the dark for 30 min and the absorbance was measured at 525 nm [89]. The ORAC method is based on the difference in the decay of fluorescein between the blank and the sample. Samples were read at 493 nm as excitation λex and 515 nm as emission λem (Varioskan Lux, Thermo Fisher Scientific, Waltham, MA, USA). In total, 1 × 10^−2^ M of fluorescein solutions in PBS (75 mM) 0.6 M AAPH in PBS (75 mM) were prepared. The sample contained 21 μL fluorescein, 2,899 μL PBS, 30 μL of the tested extract, and 50 μL AAPH [90].

### 4.3. Cell Culture

The human colon adenocarcinoma HT-29 cell line was obtained from initial plate cultures (ATCC, Manassas, VI, USA). DMEM culture medium containing glucose (4.5 g/L) and L-glutamine was used for maintenance. The medium was supplemented with 10% fetal bovine serum (FBS) and 1% penicillin-streptomycin. Cells were incubated at 37 °C in an atmosphere containing 5% CO_2_ and 95% humidity. Subcultures were performed under sterile conditions in a laminar flow hood. Cells were cultured with 90% confluence by trypsinization, for 5 min at 37 °C. Then, the trypsin was inactivated with culture medium supplemented with 10% FBS, and the cell solution was centrifuged at 1500× *g* by 3 min (Hermle Z323 K, Hermle Labortechnik GmbH, Wehingen, Germany). The cell pellet was resuspended in 1 mL of culture medium count the cells using a Neubauer chamber.

#### 4.3.1. Treatments

For the different experiments, the treatments included naringenin, BPA, naringenin with BPA, extract from the fermentation of naringenin (FEN), FEN with BPA, and the blank of the fermentation (10%), which corresponds to the culture medium required for the bacterial fermentation process. Naringenin was dissolved by stirring (SP131635Q ceramics hotirrert plate, Cimarec, Thermo Fisher Scientific, Waltham, MA, USA) for 24 h in DMEM medium supplemented with 2% FBS and 1% penicillin–streptomycin. BPA was diluted in 0.01% dimethyl sulfoxide (DMSO) and mixed with DMEM medium supplemented with 2% FBS and 1% penicillin-streptomycin. FEN was omitted using PVDF membrane filters with a pore size of 0.45 um pore sizes (MILFSLHV033RB syringe filters, Millex, Merck KGaA, Darmstadt, Germany), and mixed with DMEM medium supplemented with 2% FBS and 1% penicillin-streptomycin. For the co-exposure, both compounds were administered simultaneously. For all treatments, cells were exposed for 24 h.

#### 4.3.2. Cell Culture and Viability Test

Cell viability was determined by the MTT assay. Briefly, the cells were seeded in 96-well plates (1 × 10^4^ cells/well) and grown for 24 h in DMEM supplemented with 10% FBS. Then, the culture medium was discarded and replaced by the different treatments, consisting of naringenin (50–500 µM), FEN (25–43%), 4.4 µM BPA diluted in 0.01% dimethyl sulfoxide (DMSO) and all dissolved in DMEM supplemented with 2% FBS. The BPA concentration used in this study was selected based on the U.S. Environmental Protection Reference Dose for Chronic Oral BPA Exposure (50 µg/ kg body weight/day), considering an average body weight of 70 kg and a total water intake of 3 l [91]. For co-exposure, the IC50 of naringenin plus 4.4 µM BPA and the IC50 of FEN plus 4.4 µM BPA were used. All treatments were incubated for 24 h. After incubation, the medium was removed and a MTT solution containing 0.5 mg/mL 3-(4,5-dimethylthiazol-2-yl)-2,5-diphenyltetrazolium bromide dissolved in DMEM was added and incubated at 37 °C for 60 min. The solution was removed, and the formazan crystals were dissolved in DMSO at room temperature for 5 min. The absorbance of each sample was measured using a Multiskan Ascent (Thermo Fisher Scientific, Waltham, MA, USA, 51118307) flash spectral scanner at 540 nm. Measurements from cells without treatment were used for normalization. All experiments were performed three times each in triplicate.

#### 4.3.3. Detection of Apoptosis

Apoptosis was detected by flow cytometric analysis using Annexin-V, which identifies the externalization of phosphatidylserine. For analysis, cells were seeded in 6-well plates (4 × 10^6^ cells per 2 mL per well) in DMEM medium containing 10% FBS. After 24 h of incubation, cells were treated with naringenin (250 µM, equivalent to IC_50_), BPA (4.4 µM), FEN (37%, equivalent to IC_50_) or in co-exposure. After 24 h treatment, floating cells in the culture medium were separated, while adherent cells were collected by trypsinization. The two cell populations were then pooled and centrifuged (778× *g*, for 5 min, at 4 °C). The supernatant was discarded and 100 µL of Anexin-V was taken, which was prepared under the supplier’s conditions along with 100 µL of resuspended cells in DMEM medium supplemented with 1% SFB. They were incubated for 20 min at room temperature and analyzed in a Merck Muse Cell Analyzer (Merck KGaA, Darmstadt, Germany) [92]. 

#### 4.3.4. Cellular Necrosis 

To determine cellular necrosis, LDH activity was measured. For this purpose, cells were seeded in 96-well microplates (1 × 10^4^ cells/well) in DMEM culture medium with added FBS (10%). After 24 h of incubation, the medium was replaced by DMEM supplemented with 2% FBS containing the different concentrations of the evaluated treatments and incubated for 24 h. Cells without treatments were used as negative control and cells treated with Triton X-100 and maintained with DMEM medium containing 2% FBS were used as positive control. After 24 h of incubation, the supernatant was collected and transferred to a 96-well microplate, where 100 µL of the reaction mixture was then added according to the method recommended by the supplier (Sigma-Aldrich, St. Louis, MO, USA, MAK066), and incubated at room temperature for 20 min protected from light. Finally, absorbance was measured at 492 nm (Multiskan Ascent 51118307, Thermo Fisher Scientific, Waltham, MA, USA). 

#### 4.3.5. Superoxide Dismutase (SOD) Activity 

HT-29 cells with the different treatments were rinsed with PBS, and the cell sediments were sonicated. SOD activity was measured using the superoxide dismutase assay kit (19–160, Sigma-Aldrich, St. Louis, MO, USA), according to the manufacturer’s protocol. Absorbance was measured at 440 nm in the microplate reader. The SOD assay uses tetrazolium salt to detect superoxide radicals generated by xanthine oxidase (XO). The activity of SOD is expressed as the percentage inhibition of OX.

#### 4.3.6. Quantification of Reduced Glutathione (GSH) 

Reduced glutathione was quantified using the kit #CS0260 (Sigma-Aldrich, St. Louis, MO, USA), which provides the 5-sulfosalicylic acid needed to deproteinize the biological sample. For this technique, 1 × 10^8^ cells were seeded, washed with PBS after the treatments, later recovered and centrifuged (600× *g*, for 10 min). 5-sulfosalicylic solution equivalent to 3 times the volume of sediment was added, then frozen in liquid nitrogen and thawed in a water bath 37 °C. After centrifugation of the cells (10,000× *g*, 10 min) (Z323 K, Hermle Labortechnik GmbH, Wehingen, Germany), the supernatant was transferred to 90-well plates, where the working mixture was added (95 mM potassium phosphate buffer pH = 7, 0.95 mM EDTA, 48mM NADPH, 0.115 units/mL reduced glutathione and 0.24% 5-sulfosalicylic). Finally, absorbance was measured at 412 nm. Calculations were performed based on the previously established GSH standard curve.

### 4.4. Gene Expression 

Expression of ERβ and GPR30 genes was determined by qPCR using β-actin as housekeeping gene for normalization. Extracted RNA from treated and untreated HT-29 cells was performed using silica-gel columns (Jena, Germany). Subsequently, cDNA was synthesized using oligo dT primers and the SCRIPT cDNA kit under the following conditions: 50 °C for 40 min, 70 °C for 10 min. For qPCR, the RadiantTM Green Hi-ROX qPCR kit (Thermo Fisher Scientific, Waltham, MA, USA) was used under the following conditions: 95 °C for 2 min, 95 °C for 5 s, alignment time (Appendix A) for 20 s, and 65 °C for 10 s.

Subsequently, the expression of 84 genes was assessed using the Human RT2 Profiler Real Time PCR Array (PAHS-027A, Qiagen, Germantown, MD, USA) according to the manufacturer’s user manual. The array includes genes related to p53. Data were analyzed using RT^2^ Profiler PCR Data Analysis software from GeneGlobe (Qiagen, Germantown, MD, USA), based on the ΔΔC t method with normalization of raw data to housekeeping genes (β-actin and GAPDH). We considered sequences as potential target genes if the change between the control group and treatments was more than 2-fold (up- or down-regulated genes).

### 4.5. miRNAs Expression

For miR-200c and miR-141 expression, 2 µL of RNA (100 ng/µL) was collected and cDNA was synthesized using Stem Loop for miR-200c and miR-141 as primer and SCRIPT cDNA Synthesis Kit under the following conditions: 50 °C for 40 min, 70 °C for 10 min; then, qPCR was performed using universal antisense and specific sense primer for miR-200c and miR-141, U6 was taken as the constitutive gene synthesized using the oligo dT primer and the corresponding FWD and REV primers for qPCR (StepOne™ 48-well, Thermo Fisher Scientific, Waltham, MA, USA) (Appendix A). The qPCR conditions were as follows: 95 °C for 2 min, 95 °C for 5 s, 60 °C for 20 s, and 65 °C for 10 s using the RadiantTM Green Hi-ROX qPCR kit (QS2100, Alkali Scientific Inc., FL, USA) The predictive power of suppression of protein-coding gene expression was evaluated using miRmap software, Swiss Institute of Bioinformatcs, Université De Genève. 

### 4.6. Statistical Analysis 

The results obtained were analyzed using a two-way and one-way variance test (ANOVA), followed by a *post hoc* Tukey test to compare the treatment and control groups. Analyzes were performed using the Prism 8 program (GraphPad Software, San Diego, CA, USA).

## 5. Conclusions

The results of the present study showed that microbial metabolism during fermentation of naringenin in the colon produced by-products of biological interest such as apigenin and 3HPP, and that both naringenin and FEN dose-dependently reduced the viability of HT-29 colon cancer cells, both alone and with BPA co-exposure. Naringenin promoted extrinsic apoptosis through overexpression of TNFRST10D/CRADD/CASP-2 and intrinsic apoptosis through PTEN/BBC3/APAF-1/CASP-9 on HT-29 cells (Figure 9). FEN promoted apoptotic mechanisms through intrinsic pathways likely induced by H_2_O_2_ generated by SOD activity (Figure 10). In contrast, BPA decreased the expression of BBC3, an important gene that promotes apoptosis, decreases tumour suppressor genes such as PTEN and also decreases MLH-1, which plays a role in DNA repair related to cell proliferation. In addition, BPA-mediated overexpression of miR-200c is thought to be related to the down-regulation of PTEN, which regulates the cell cycle and promotes apoptosis on HT-29 cells. Furthermore, simultaneous exposure of naringenin and FEN to BPA triggers overexpression of the TNF gene, which causes overexpression of the FASL gene, mediating activation of apoptosis via the extrinsic pathway (Figure 11).

The decrease in cell viability as well as the recovery of ERβ gene expression on HT-29 cells under treatment with naringenin and its extract FEN suggest that the expression of this receptor is associated with apoptotic mechanisms and DNA damage repair, although the increase in ERα gene expression related to proliferation processes, increased ERβ expression plays a very important role in inhibiting colon cancer development. Finally, ERβ could modulate the low expression of miR-141 targeting PTEN in HT-29 cells treated with FEN. 

Overall, our results demonstrated that naringenin and its fermentation extract activate apoptotic signalling pathways even in the presence of BPA, suggesting the effect of naringenin in inhibiting colon cancer development.

Finally, this provides an important perspective for further exploration of naringenin as an agent with enhanced efficacy against the identified molecular targets given the high BPA burden worldwide.

## Figures and Tables

**Figure 1 molecules-27-06588-f001:**
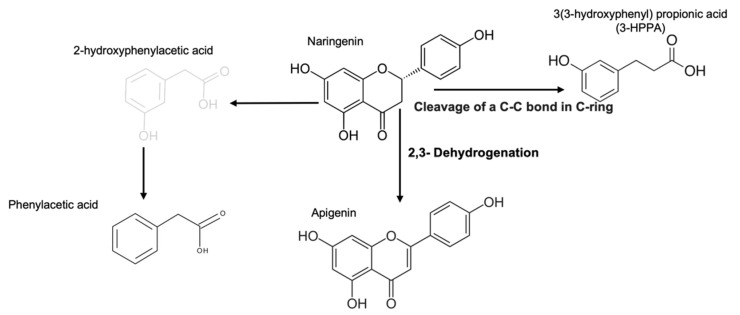
Proposed metabolic pathway of naringenin by colonic microbiota. Detected metabolites are in black, non-detected metabolites are in grey.

**Figure 2 molecules-27-06588-f002:**
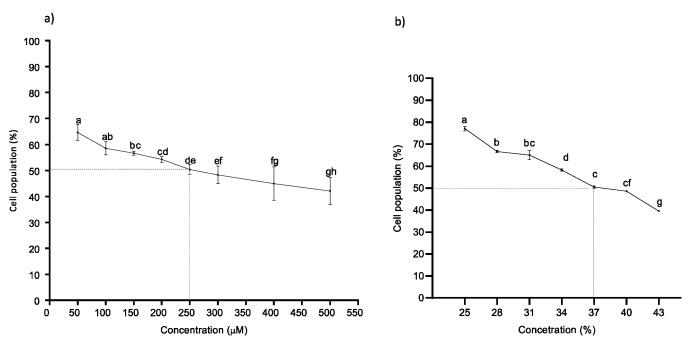
Cell viability of HT-29 cells after treatment with naringenin or FEN. The effect of (**a**) naringenin (50–500 µM) and of (**b**) FEN (25–43%) was evaluated on HT-29 cells after 24 h post incubation (MTT assay). Viability is expressed as a percentage of cellular activity between the exposed group to naringenin (**a**) and FEN (**b**). The experimental data are expressed as the mean value ± SD of the three independent experiments with three replicates each one. Letters indicate statistical difference (*p* < 0.05).

**Figure 3 molecules-27-06588-f003:**
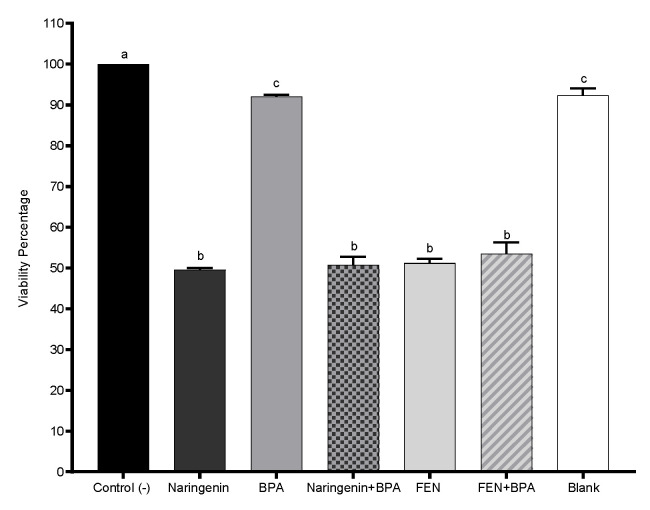
The effect of naringenin (250 µM), BPA (4.4 µM), FEN (37%), co-treatment naringenin and FEN with BPA, and blank of fermentation (10%) on HT-29 cells viability was evaluated after 24 h incubation by MTT assay. The experimental data are expressed as the mean value ± SD of the three independent experiments with three replicates each one. Letters indicate statistical difference (*p* < 0.05).

**Figure 4 molecules-27-06588-f004:**
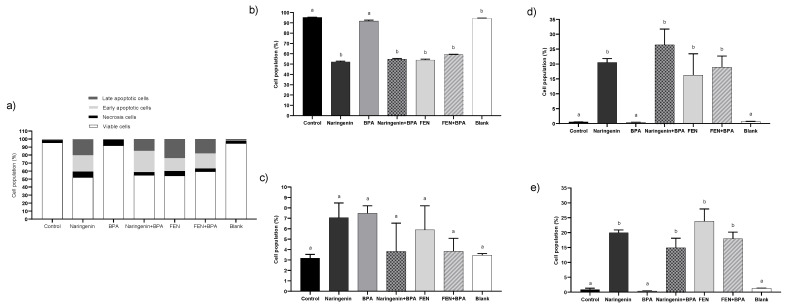
Flow cytometry analysis with Annexin V. HT-29 cells were incubated 24 h with the treatments: 250 µM naringenin, 4.4 µM BPA, 37% FEN, co-treatment naringenin + BPA or co-treatment FEN + BPA, and blank of fermentation (10%) on. (**a**) All cells (**b**) Viable cells, (**c**) Necrosis cells, (**d**) Early apoptotic cells, (**e**) Late apoptotic cells. The experimental data are expressed as the mean value ± SD of the two independent experiments with two replicates each one. Letters indicate statistical difference (*p* < 0.05).

**Figure 5 molecules-27-06588-f005:**
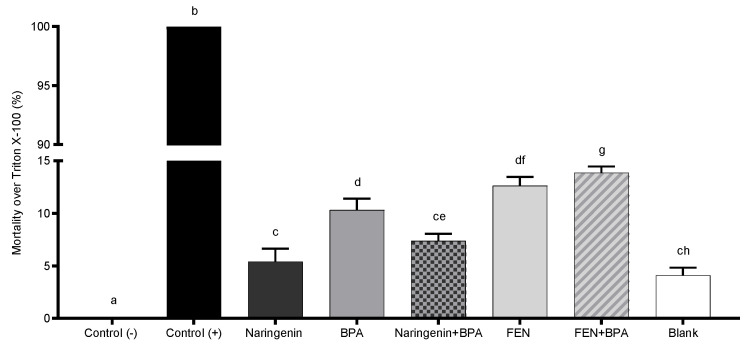
The effect of 250 µM naringenin, 4.4 µM BPA, 37% FEN, co-treatment naringenin + BPA or co-treatment FEN + BPA, and blank of fermentation (10%) on HT-29 cells was evaluated after 24 h incubation by LDH cytotoxicity assay. The total LDH controls can also be used to calculate the total effect of a specific condition. The experimental data are expressed as the mean value ± SD of the three independent experiments with three replicates of each one. Letters indicate statistical difference (*p* < 0.05).

**Figure 6 molecules-27-06588-f006:**
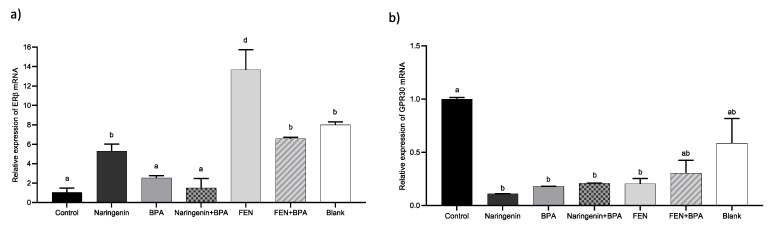
Relative mRNA expression of ERβ (**a**) and GPR30 (**b**) in HT-29 cells treated with250 µM naringenin, 4.4 µM BPA, 37% FEN, co-treatment naringenin + BPA or co-treatment FEN + BPA and blank of fermentation (10%). Data expression was normalized with β-actin. The experimental data are expressed as the mean value ± SD of the two independent experiments. Letters indicate statistical difference (*p* < 0.05).

**Figure 7 molecules-27-06588-f007:**
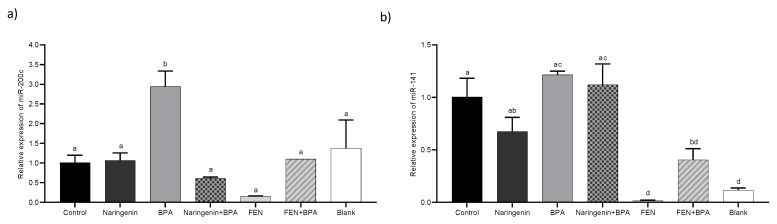
Relative miRNA expression; (**a**) miR-200c, (**b**) miR-141 relative to U6 in HT-29 cells treated with 250 µM naringenin, 4.4 µM BPA, 37% FEN, co-treatment naringenin + BPA or co-treatment FEN + BPA. The experimental data are expressed as the mean value ± SD of the two independent experiments. Letters indicate statistical difference by Tukey–Kramer’s Test (*p* < 0.05).

**Figure 8 molecules-27-06588-f008:**
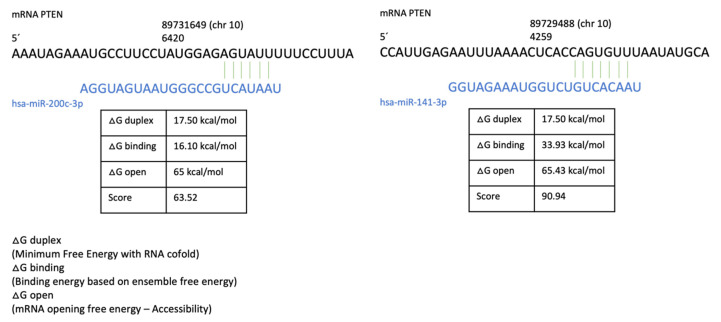
Comprehensive prediction of PETEN repression by miR-200c and miR-141 simulated in miRmap.

**Figure 9 molecules-27-06588-f009:**
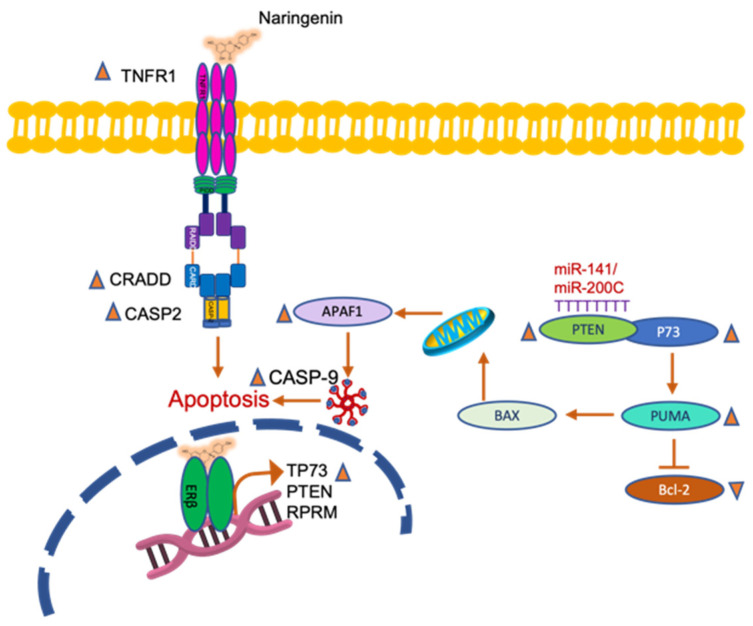
Extrinsic and intrinsic apoptosis by naringenin treatment.

**Figure 10 molecules-27-06588-f010:**
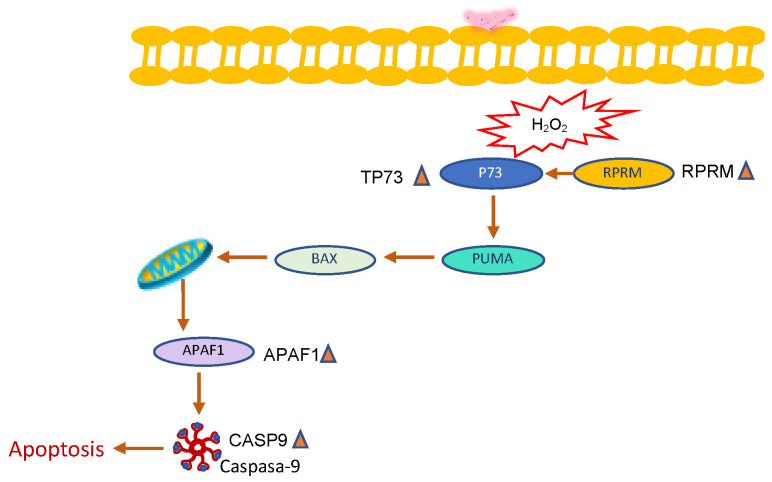
Intrinsic apoptosis by FEN treatment.

**Figure 11 molecules-27-06588-f011:**
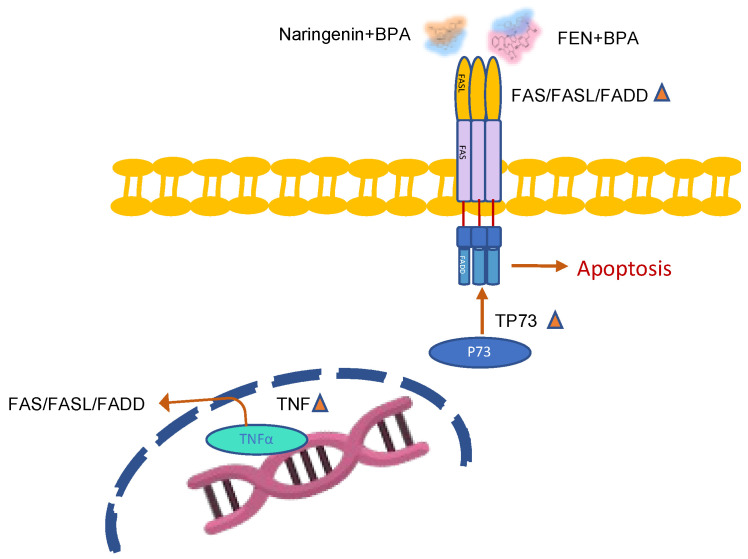
Extrinsic apoptosis by co-exposition treatment, naringenin +BPA and FEN+BPA.

**Table 1 molecules-27-06588-t001:** Identification of FEN compounds and Naringenin by UPLC-MS and their antioxidant capacity.

UPLC-MS				Antioxidant Capacity	
Treatment	Compounds	Retention Time	Molar Mass g/mol	DPPH µM eq. Trolox	ORAC µM eq. Trolox
Blank	Phenylacetic acid	6.50 ± 0.00	135.04	35.27 ± 0.70 ^a^	2.02 ± 0.10 ^a^
	Secoisolariciresinol	11.00 ± 0.00	361.16		
FEN	Naringenin	8.63 ± 0.01	271.06	52.97 ± 1.40 ^b^	22.12 ± 6.70 ^b^
	3-HPPA *	4.93 ± 0.03	167.03		
	Apigenin	8.63 ± 0.01	271.06		
	Secoisolariciresinol	11.60 ± 0.00	361.16		
	Phenylacetic acid	6.43 ± 0.01	135.94		
Naringenin	Naringenin	8.63 ± 0.00	271.06	92.37 ± 1.10 ^c^	22.31 ± 6.80 ^b^

* 3-HPPA. 3-(3-hydroxyphenyl)propionic acid). Results are expressed as the average of three independent experiments with three replicates ± SD. Letters indicate statistical difference (*p* < 0.05). Trolox equivalent antioxidant capacity (TEAC).

**Table 2 molecules-27-06588-t002:** Effect of Naringenin, BPA, FEN and co-exposition treatments on oxidative stress in HT29 cells after 24 h of treatment.

	GSH nmol GSH/mL	SOD % Inhibition of O_2_
Control	160.22 ± 2.4 d	1.62 ± 1.1 a
Naringenin	110.75 ± 1.5 a	3.05 ± 0.3 a
BPA	140.03 ± 4.1 c	6.03 ± 1.1 a
Naringenin + BPA	129.19 ± 0.9 b	5.2 ± 0.8 a
FEN	144.15 ± 5.6 d	13.18 ± 2 b
FEN + BPA	157.99 ± 1.5 c	4.8 ± 1.0 a
Blank	152.67 ± 0.4 d	3.2 ± 0.4 a

Results are expressed as the average of three independent experiments with three replicates ± SD. Different letters by columns express significant differences by Tukey–Kramer’s Test (*p* < 0.05). Reduced glutathione; SOD. Superoxide dismutase.

**Table 3 molecules-27-06588-t003:** Fold-change in the expression levels of related genes to the p53 signaling cascade in HT-29 cells treated with 250 µM naringenin, 4.4 µM BPA, 37% FEN, co-treatment naringenin + BPA or co-treatment FEN + BPA in relation to negative control cells (up-regulation +, down-regulation −).

Symbol	Description	Naringenin	BPA	Naringenin + BPA	FEN	FEN + BPA
APAF1	Apoptotic peptidase activating factor 1 (Other apoptosis genes)	1.11	1.52	21.10	2.62	27.86
ATM	Ataxia telangiectasia mutated (Negative regulation of the cell cycle)	1.00	1.02	3.13	1.12	8.66
ATR	Ataxia telangiectasia and Rad3-related (Cell cycle checkpoint)	−1.20	−2.16	3.28	−1.07	5.63
ADGRB1	Brain-specific angiogenesis inhibitor 1 (Other genes related to inhibition proliferation)	−1.11	−1.04	16.54	2.39	16.04
BAX	BCL2-associated X protein (Induction of apoptosis)	−1.27	−1.41	1.80	−1.45	5.06
BBC3	BCL2 binding component 3 (Induction of apoptosis)	1.46	−4.96	13.67	1.76	6.87
BCL2	B-cell CLL/lymphoma 2 (Anti-apoptosis)	−2.17	−1.21	40.54	1.70	146.89
BCL2A1	BCL2-related protein A1 (Anti-apoptosis)	1.01	−1.21	19.29	3.76	43.36
BID	BH3 interacting domain death agonist (Induction of apoptosis)	1.52	−1.16	1.79	1.11	1.17
BIRC5	Baculoviral IAP repeat containing 5 (Anti-apoptosis)	1.36	1.41	10.53	3.12	21.64
BRCA1	Breast cancer 1, early onset (Cell cycle checkpoint)	1.11	1.15	4.04	1.27	4.29
BRCA2	Breast cancer 2, early onset (Regulation of the cell cycle)	3.28	−1.34	62.42	1.92	19.02
BTG2	BTG family, member 2 (Other genes related to inhibition proliferation)	−1.24	−1.56	15.30	−2.66	2.88
CASP2	Caspase 2, apoptosis-related cysteine peptidase (Anti-apoptosis, apoptosis)	2.14	8.11	2.99	−1.58	4.30
CASP9	Caspase 9, apoptosis-related cysteine peptidase (Other apoptosis genes)	3.99	−1.06	2.36	2.03	12.27
CCNB1	Cyclin B1 (Other cell cycle genes)	1.32	1.30	1.47	1.46	2.12
CCNE1	Cyclin E1 (Cell cycle checkpoint)	1.11	1.13	3.96	1.88	8.23
CCNG1	Cyclin G1 (Cell cycle checkpoint)	1.02	1.02	−3.41	−1.49	1.22
CCNH	Cyclin H (Regulation of the cell cycle)	1.31	−1.04	1.76	1.69	9.19
CDC25A	Cell division cycle 25 homolog A (S. pombe) (Regulation of the cell cycle)	1.27	−1.11	18.49	2.02	12.59
CDC25C	Cell division cycle 25 homolog C (S. pombe) (Cell proliferation)	1.45	1.14	4.31	1.73	7.16
CDK1	Cyclin-dependent kinase 1 (Regulation of the cell cycle)	1.17	1.30	−2.86	1.34	3.73
CDK4	Cyclin-dependent kinase 4 (Regulation of the cell cycle)	1.13	−1.39	1.84	−1.14	1.23
CDKN1A	Cyclin-dependent kinase inhibitor 1A (p21, Cip1) (Cell cycle arrest)	1.36	−1.13	3.37	5.85	48.32
CDKN2A	Cyclin-dependent kinase inhibitor 2A (melanoma, p16, inhibits CDK4) (Cell cycle arrest)	1.38	1.11	2.18	1.44	1.62
CHEK1	CHK1 checkpoint homolog (S. pombe) (Cell cycle arrest)	1.30	1.27	2.31	1.24	4.75
CHEK2	CHK2 checkpoint homolog (S. pombe) (Cell cycle arrest)	1.54	1.18	2.74	1.74	7.30
CRADD	CASP2 and RIPK1 domain containing adaptor with death domain (Induction of apoptosis)	2.96	−1.48	56.18	1.38	10.58
DNMT1	DNA (cytosine-5-)-methyltransferase 1 (DNA repair genes)	−1.05	1.01	2.20	1.21	2.07
E2F1	E2F transcription factor 1 (Regulation of the cell cycle)	1.29	1.03	3.59	1.39	6.35
E2F3	E2F transcription factor 3 (Regulation of the cell cycle)	1.18	−1.01	6.17	−1.44	5.39
EGFR	Epidermal growth factor receptor (Other genes related to cell growth, proliferation, and differentiation)	1.11	−1.05	4.07	1.04	8.01
EGR1	Early growth response 1 (Other genes related to cell growth, proliferation, and differentiation)	1.44	−1.06	22.88	−1.00	15.41
EI24	Etoposide induced 2.4 mRNA (Induction of apoptosis)	1.17	−1.12	−1.44	1.06	1.62
ESR1	Estrogen receptor 1 (Cell growth and differentiation)	1.11	3.63	63.17	5.11	33.01
FADD	Fas (TNFRSF6)-associated via death domain (Induction of apoptosis)	1.41	−1.07	1.19	−1.05	2.05
FAS	Fas (TNF receptor superfamily, member 6) (Induction of apoptosis)	1.01	1.43	12.09	2.13	15.30
FASLG	Fas ligand (TNF superfamily, member 6) (Induction of apoptosis)	1.86	−1.31	35.45	4.63	52.37
FOXO3	Forkhead box O3 (Other genes related to cell growth, proliferation and differentiation)	1.29	1.01	1.69	1.15	4.45
GADD45A	Growth arrest and DNA-damage-inducible, alpha (Other apoptosis genes)	−1.16	−1.13	4.97	2.92	4.79
GML	Glycosylphosphatidylinositol anchored molecule like (Other genes related to cell growth, proliferation and differentiation) protein (Other apoptosis genes)	1.95	1.05	37.20	4.98	60.37
HDAC1	Histone deacetylase 1 (Anti-apoptosis)	1.49	−1.01	−1.57	−1.05	1.64
HK2	Hexokinase 2 (Regulation of the cell cycle)	1.24	1.62	3.55	2.79	9.41
IGF1R	Insulin-like growth factor 1 receptor (Regulation of the cell cycle)	1.27	1.17	3.50	−1.05	5.58
IL6	Interleukin 6 (interferon, beta 2) (Positive regulation of the cell proliferation)	−1.40	−1.01	13.63	3.45	30.56
JUN	Jun proto-oncogene (Other genes related to cell growth, proliferation and differentiation)	−2.06	−1.72	1.57	−3.60	1.59
KAT2B	K(lysine) acetyltransferase 2B (Other genes related to cell growth, proliferation and differentiation)	1.18	1.04	22.40	2.51	16.65
KRAS	V-Ki-ras2 Kirsten rat sarcoma viral oncogene homolog (Other genes related to cell growth, proliferation)	−1.10	−1.01	−1.39	−1.42	2.19
MCL1	Myeloid cell leukemia sequence 1 (BCL2-related) (Anti-apoptotic)	1.12	−1.17	2.52	1.47	4.03
MDM2	Mdm2 p53 binding protein homolog (mouse) (Negative regulation of cell proliferation)	1.14	1.46	21.69	2.85	36.22
MDM4	Mdm4 p53 binding protein homolog (mouse) (Negative regulation of cell proliferation)	−1.39	1.66	3.80	−1.28	7.25
MLH1	MutL homolog 1, colon cancer, nonpolyposis type 2 (E. coli) (Other cell cycle genes)	4.48	−7.94	2.15	1.25	5.32
MSH2	MutS homolog 2, colon cancer, nonpolyposis type 1 (E. coli) (DNA repair genes)	1.39	1.07	−1.24	1.30	4.25
MYC	V-myc myelocytomatosis viral oncogene homolog (avian) (Cell cycle arrest)	−1.02	−1.21	1.25	−1.10	1.79
MYOD1	Myogenic differentiation 1 (Cell growth and differentiation)	3.99	−1.10	75.89	5.20	26.82
NF1	Neurofibromin 1 (Negative regulation of the cell cycle)	1.16	1.09	4.35	1.56	11.20
NFKB1	Nuclear factor of kappa light polypeptide gene enhancer in B-cells 1 (Anti-apoptosis)	−1.22	−1.08	6.63	−1.12	5.06
PCNA	Proliferating cell nuclear antigen (Cell proliferation)	1.14	−1.21	2.15	1.01	2.18
PIDD1	P53-induced death domain protein (Apoptosis)	1.09	1.04	12.74	1.87	10.41
PPM1D	Protein phosphatase, Mg2+/Mn2+ dependent, 1D (Negative regulation of the cell proliferation)	−1.35	1.14	10.99	1.31	6.36
PRC1	Protein regulator of cytokinesis 1 (Other cell cycle genes)	1.12	−1.49	1.41	−1.17	1.59
PRKCA	Protein kinase C, alpha (Cell proliferation)	1.75	−1.10	4.29	1.71	8.28
PTEN	Phosphatase and tensin homolog (Negative regulation of the cell cycle)	2.70	−3.46	1.20	1.72	6.98
PTTG1	Pituitary tumor-transforming 1 (DNA repair genes)	1.68	−1.38	1.74	2.72	6.99
RB1	Retinoblastoma 1 (Cell cycle checkpoint)	−1.01	−3.32	1.04	−1.08	3.02
RELA	V-rel reticuloendotheliosis viral oncogene homolog A (avian) (Cell proliferation)	1.49	−1.19	1.18	−1.19	1.87
RPRM	Reprimo, TP53 dependent G2 arrest mediator candidate (Cell cycle arrest)	3.15	−1.39	59.96	5.71	42.08
SESN2	Sestrin 2 (Negative regulation of the cell proliferation)	−1.70	−1.01	2.86	1.69	4.24
SIAH1	Seven in absentia homolog 1 (Drosophila) (Other apoptosis genes)	1.61	−1.42	1.21	1.39	4.26
SIRT1	Sirtuin 1 (Other apoptosis genes)	−1.00	−3.18	2.05	−1.09	2.70
STAT1	Signal transducer and activator of transcription 1, 91kDa (Regulation of the cell cycle)	−1.08	−4.06	1.19	−1.21	−9.20
TADA3	Transcriptional adaptor 3 (Regulation of the cell cycle)	1.39	−2.66	1.67	−1.17	−9.68
TNF	Tumor necrosis factor (Anti-apoptosis)	1.71	−1.62	32.52	3.89	47.68
TNFRSF10B	Tumor necrosis factor receptor superfamily, member 10b (Induction of apoptosis)	1.08	1.23	3.59	1.75	3.66
TNFRSF10D	Tumor necrosis factor receptor superfamily, member 10d, decoy with truncated death domain (Anti-apoptosis)	4.98	−1.13	14.79	1.74	9.71
p53	Tumor protein p53 (Induction of apoptosis, negative regulation of the cell cycle, DNA repair genes)	1.15	1.41	−1.00	−1.80	1.19
TP53AIP1	Tumor protein p53 regulated apoptosis inducing protein 1(Other apoptosis genes)	3.99	−1.10	75.89	3.62	27.66
TP53BP2	Tumor protein p53 binding protein, 2 (Other apoptosis genes)	1.35	−4.06	1.46	1.58	6.08
TP63	Tumor protein p63 (Induction of apoptosis)	2.11	−1.24	39.99	6.29	66.82
TP73	Tumor protein p73 (Induction of apoptosis)	3.99	1.12	75.89	2.65	101.66
TRAF2	TNF receptor-associated factor 2 (Cell proliferation)	1.21	−1.20	2.49	1.36	4.12
TSC1	Tuberous sclerosis 1 (Negative regulation of the cell cycle)	−1.19	1.18	5.12	1.42	−1.04
WT1	Wilms tumor 1 (Negative regulation of the cell cycle)	1.39	1.02	75.89	3.21	14.29
XRCC5	X-ray repair complementing defective repair in Chinese hamster cells 5 (double-strand-break rejoining) (DNA repair genes)	1.21	−1.31	−1.84	−1.73	−30.13

Bold numbers indicate a ±2-fold change compared to negative control. β-Actin and GAPDH were used as housekeeping genes.

## Data Availability

The data presented in this study are available in Appendix A.

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
