# Peer review of "Fermentation Extract of Naringenin Increases the Expression of Estrogenic Receptor β and Modulates Genes Related to the p53 Signalling Pathway, miR-200c and miR-141 in Human Colon Cancer Cells Exposed to BPA"

_molecules, 2022, doi:10.3390/molecules27196588_

Round 1
Reviewer 1 Report
This study examined the molecular consequences of co-exposure to FEN and BPA in HT-29 colon cancer cells. The authors assessed ER, miR-200c, miR-141, and 24 p53-related genes. The results indicated that co-exposed cells are under higher stress, which forces them to mediate other ways to apoptosis.
The authors mentioned their results in line 75-78 in the introduction section. Before this sentence, the authors declared their aim of this study, which is good and clear, but there might be no need to put result in the introduction section.
In section 2.1, line 85 mentioned several authors have found dehy- 85 droxylations, hydroxylations and deglycosylations of phenolic compounds generated by 86 microbial metabolism. Did the authors cite other researchers’ results to support their result? If so, they should add references here and I would also suggest to move to the discussion section. If not, the expression here is quite confusing.
In Section 4.2, the authors mentioned the saliva was collected from 4 volunteers. More details of those volunteers, such as whether they are healthy people or cancer patients, could be included to better describe the sample source.
For the statistical analysis, the selection of ANOVA was proper here. But the authors did not mention whether their data met the normality and homogeneity of variance assumptions, which would enable them to use the post hoc Tukey Test
Author Response
Referee: 1
- Reviewer: This study examined the molecular consequences of co-exposure to FEN and BPA in HT-29 colon cancer cells. The authors assessed ER, miR-200c, miR-141, and 24 p53-related genes. The results indicated that co-exposed cells are under higher stress, which forces them to mediate other ways to apoptosis.
- Authors’ response: We appreciate the reviewer’s comments.
- Reviewer: The authors mentioned their results in line 75-78 in the introduction section. Before this sentence, the authors declared their aim of this study, which is good and clear, but there might be no need to put result in the introduction section.
- Authors’ response: We agree with the reviewer comment. We deleted the results in the introduction section, the objective was kept.
Revised Manuscript:
Therefore, the aim of this study was to investigate the anticarcinogenic effect of naringenin and its fermented extract from colon upon simultaneous exposure to the disruptor BPA in human adenocarcinoma cells HT-29, and to elucidate the molecular mechanisms involved. (Page 2, line 72-75).
- Reviewer: In section 2.1, line 85 mentioned several authors have found dehy- 85 droxylations, hydroxylations and deglycosylations of phenolic compounds generated by 86 microbial metabolism. Did the authors cite other researchers’ results to support their result? If so, they should add references here and I would also suggest to move to the discussion section. If not, the expression here is quite confusing.
- Authors’ response: We have added the appropriate citations in the Results section; the same ones we rely on in the Discussion section.
Revised Manuscript:
…compounds generated by microbial metabolism to produce various byproducts, as we show (Figure 1). [22,23] (Page 2, line 84)
- Rechner, A. R.; Smith, M. A.; Kuhnle, G.; Gibson, G. R.; Debnam, E. S.; Srai, S. K. S.; Moore, K. P.; Rice-Evans, C. A. Colonic Metabolism of Dietary Polyphenols: Influence of Structure on Microbial Fermentation Products. Free Radic. Biol. Med 2004, 36 (2), 212–225.
- Zeng, X.; Su, W.; Zheng, Y.; He, Y.; He, Y.; Rao, H.; Peng, W.; Yao, H. Pharmacokinetics, Tissue Distribution, Metabolism, and Excretion of Naringin in Aged Rats. Front. Pharmacol 2019, 10-34.
- Reviewer: In Section 4.2, the authors mentioned the saliva was collected from 4 volunteers. More details of those volunteers, such as whether they are healthy people or cancer patients, could be included to better describe the sample source.
- Authors’ response: Thanks, corrected. We have included more details of the volunteers.
Revised Manuscript:
…, healthy, with no chronic degenerative diseases or contact infections in the past 30 days, no use of antibiotics, medications in general, or alcohol in the past 20 days. Subjects had to be of legal age and not eating a vegetarian or keto diet. The sample was diluted with 10 mL of distilled water. (Page 18, lines 537-540).
- Reviewer: For the statistical analysis, the selection of ANOVA was proper here. But the authors did not mention whether their data met the normality and homogeneity of variance assumptions, which would enable them to use the post hoc Tukey Test
- Authors’ response: We appreciate the reviewer comment. We did not perform a normality test, because normality tests on a small number of data groups, can lead to erroneous results, according to other authors. Läärä (2009) gives several reasons for not using the preliminary normality test, including small samples, because the power of the test is low. Tukey–Kramer multiple comparison test has been applied by several authors to analyze the statistically significant difference in cell cultures, e.g.: Imam et al., (2021), Kuwahara et al., (2022), Komine-Aizawa et al., (2020), among others.
Läärä, E. Statistics: reasoning on uncertainty, and the insignificance of testing null. Ann. Zool. Fennici 2009, 46: 138–157. https://doi.org/10.5735/086.046.0206
Imam, S. S., Alshehri, S., Altamimi, M. A., Hussain, A., Qamar, W., Gilani, S. J., Zafar, A., Alruwaili, N.K., Alanazi, S., Almutairy, B. K. Formulation of Piperine–Chitosan-Coated Liposomes: Characterization and In Vitro Cytotoxic Evaluation. Molecules 2021, 26(11), 3281. doi:10.3390/molecules26113281
Kuwahara, M., Akasaki, Y., Goto, N., Kurakazu, I., Sueishi, T., Toya, M., Uchida, T., Tsutsui, T., Hirose, R., Tsushima, H., Nakashima, Y. Fluvastatin promotes chondrogenic diferentiation of adipose-derived mesenchymal stem cells by inducing bone morphogenetic protein 2. BMC Pharmacology and Toxicology 2022, 23:61. https://doi.org/10.1186/s40360-022-00600-7
Komine-Aizawa, S., Aizawa, S., Takano, C., Hayakawa, S. Interleukin-22 promotes the migration and invasion of oral squamous cell carcinoma cells. Immunological Medicine 2020, 1–9. doi:10.1080/25785826.2020.1775060

Reviewer 2 Report
The paper "Fermentation extract of naringenin increases the expression of estrogenic receptor β and modulates genes related to the p53 signalling pathway, miR-200c and miR-141 in human colon cancer cells exposed to BPA" by Castaneda et al. shows that microbial metabolism during fermenta- 698 tion of naringenin in the colon produced by-products of biological interest such as apig- 699 enin and 3HPP, and that both naringenin and FEN dose-dependently reduced the viabil- 700 ity of HT-29 colon cancer cells, both alone and with BPA co-exposure. The work is interesting, anyway I have some comments to do.Major suggestions.
- The authors determined the antioxidant capacity of FEN and naringenin by DPPH and ORAC assay. Although the DPPH is a synthetic radical it is widely used to compare the radical scavenging activity of compounds/extracts. The authors discussed only the results obtained from the ORAC, but the different result obtained from the DPPH should be discussed. It could be interesting because the ORAC measures the molecule ability to transfer hydrogen atoms to oxygen radicals (doi.org/10.1021/jp307746c), while DPPH measures the electron transfer reaction between DPPH radical and antioxidant (DOI:10.3390/antiox10081224). Overall, the FEN showed a different antioxidant capacity in comparison with the naringenin.
- The authors claim that “On the other hand, GSH, the most potent antioxidant in the body, was significantly decreased by naringenin and naringenin… suggesting that GSH, as an electron donor, undergoes oxidation- reduction reactions to neutralize free radicals generated by oxidative stress in the cell.” It is not a clear sentence. First, it is known that several flavonoids, including naringenin, induce GSH depletion but through activation of multidrug resistance protein 1 (MRP1), an ATP-binding cassette (ABC) transporter and that this could lead to sensitize cancer cells (doi: 10.1016/j.freeradbiomed.2006.03.002). Please explain and discuss better this result.
Minor suggestions.
- Line 206. In the sentence “Another gene involved in apoptosis is caspase-2 and CRADD”. Please add a references.
- Line 210. Please add a reference relative to the role of PTEN in the cancer modulation (i.e. doi: 10.3390/genes11070719; doi: 10.3389/fonc.2015.00024).
- Line 316. In the sentence “Antioxidant capacity is one of the strategies by which flavonoids can inhibit cancer cells proliferation.” Generally, flavonoids inhibit cancer cell growth by interaction with overexpressed or mutated proteins (DOI:10.3390/ijms222111833; https://doi.org/10.3389/fphar.2021.639840) or by acting as pro-oxidative agents (doi: 10.3390/nu12020457). Please correct and better discuss the previous sentence.
Author Response
Referee: 2
- Reviewer: The paper "Fermentation extract of naringenin increases the expression of estrogenic receptor β and modulates genes related to the p53 signalling pathway, miR-200c and miR-141 in human colon cancer cells exposed to BPA" by Castaneda et al. shows that microbial metabolism during fermenta- 698 tion of naringenin in the colon produced by-products of biological interest such as apig- 699 enin and 3HPP, and that both naringenin and FEN dose-dependently reduced the viabil- 700 ity of HT-29 colon cancer cells, both alone and with BPA co-exposure. The work is interesting, anyway I have some comments to do.
- Authors’ response: We appreciate the reviewer’s comments.
- Reviewer: The authors determined the antioxidant capacity of FEN and naringenin by DPPH and ORAC assay. Although the DPPH is a synthetic radical it is widely used to compare the radical scavenging activity of compounds/extracts. The authors discussed only the results obtained from the ORAC, but the different result obtained from the DPPH should be discussed. It could be interesting because the ORAC measures the molecule ability to transfer hydrogen atoms to oxygen radicals (doi.org/10.1021/jp307746c), while DPPH measures the electron transfer reaction between DPPH radical and antioxidant (DOI:10.3390/antiox10081224). Overall, the FEN showed a different antioxidant capacity in comparison with the naringenin.
- Authors’ response: Thanks for the comment. We agree with the reviewer regarding the observations on ORAC and DPPH behavior, so we have expanded the discussion.
Revised Manuscript:
FEN exhibited a lower antioxidant capacity than naringenin as measured by DPPH. This is probably because the by-products found in FEN differ from naringenin in the amount and position of the hydroxyl groups. Apigenin, a byproduct of FEN, has a higher antioxidant potential than naringenin because of the 2,3-dehydrogenation reaction leading to conjugative C=C binding. However, the interaction of different metabolites leads to antagonism that reduces the ability to donate electrons to the DPPH radical, which is the fundamental basis of the assay. Second, in the evaluation of FEN and naringenin by ORAC method, which is based on the ability of the molecule to transfer hydrogen atoms to oxygen radicals, there is no difference between FEN and naringenin. [28,29] Thus, it can be seen that naringenin has a great ability to transfer electrons and has the same ability to transfer hydrogen atoms to oxygen radicals as the group of by-products formed after fermentation. (Page 14, lines 309-319).
- Cai, Y.; Li, X.; Shen, P.; Zhang, D. CCAT2 Is an Oncogenic Long Non-Coding RNA in Pancreatic Ductal Adenocarcinoma. Biol Res 2018, 51 (1), 1.
- Heim, K. E., Tagliaferro, A. R., & Bobilya, D. J. Flavonoid antioxidants: Chemistry, metabolism and structure-activity relationships. J Nutr Biochem 2002, 13(10), 572-584.
- Reviewer: The authors claim that “On the other hand, GSH, the most potent antioxidant in the body, was significantly decreased by naringenin and naringenin… suggesting that GSH, as an electron donor, undergoes oxidation- reduction reactions to neutralize free radicals generated by oxidative stress in the cell.” It is not a clear sentence. First, it is known that several flavonoids, including naringenin, induce GSH depletion but through activation of multidrug resistance protein 1 (MRP1), an ATP-binding cassette (ABC) transporter and that this could lead to sensitize cancer cells (doi: 10.1016/j.freeradbiomed.2006.03.002). Please explain and discuss better this result.
- Authors’ response: We appreciate the reviewer's comment. To enrich the discussion, we have attempted to include the comments.
Revised Manuscript:
Another antioxidant system is GSH. Although it is one of the most potent antioxidant systems, large amounts of GSH have been associated with metastasis and chemoresistance in cancer, as an adaptive response of the cancer cell to large amounts of free radicals. We hypothesize that GSH depletion after naringenin treatment, as reported by other authors, is mediated by multidrug resistance protein 1 (MRP1), which promotes increased sensitivity of cancer cells. [6,37-39] (Page 15, lines 347-352).
- Kopustinskiene, D.M; Jakstas, V.; Savickas, A.; Bernatoniene, J. Flavonoids as Anticancer Agents. Nutrients. 2020 12(2), 457.
- Reviewer: Line 206. In the sentence “Another gene involved in apoptosis is caspase-2 and CRADD”. Please add a references.
- Authors’ response: We have added the relevant reference.
Revised Manuscript:
Another gene involved in apoptosis is caspase-2 and CRADD. [24] (Page 7, line 205).
- Reviewer: Line 210. Please add a reference relative to the role of PTEN in the cancer modulation (i.e. doi: 10.3390/genes11070719; doi: 10.3389/fonc.2015.00024).
- Authors’ response: We consider relevant suggested references.
Revised Manuscript:
PTEN is involved in the modulation of several cancer processes, including apoptosis. [25,26] (Page 7, line 210).
- Milella, M.; Falcone, I.; Conciatori, F.; Cesta Incani, U.; Del Curatolo, A.; Inzerilli, N.; Nuzzo, C. M. A.; Vaccaro, V.; Vari, S.; Cognetti, F.; Ciuffreda, L. PTEN: Multiple Functions in Human Malignant Tumors. Oncol. 2015, 5.
- Fusco, N.; Sajjadi, E.; Venetis, K.; Gaudioso, G.; Lopez, G.; Corti, C.; Guerini Rocco, E.; Criscitiello, C.; Malapelle, U.; Invernizzi, M. PTEN Alterations and Their Role in Cancer Management: Are We Making Headway on Precision Medicine? Genes 2020, 11 (7), 719.
- Reviewer: Line 316. In the sentence “Antioxidant capacity is one of the strategies by which flavonoids can inhibit cancer cells proliferation.” Generally, flavonoids inhibit cancer cell growth by interaction with overexpressed or mutated proteins (DOI:10.3390/ijms222111833; https://doi.org/10.3389/fphar.2021.639840) or by acting as pro-oxidative agents (doi: 10.3390/nu12020457). Please correct and better discuss the previous sentence.
- Authors’ response: We appreciate the reviewer's comment. We have suggested a better explanation.
Revised Manuscript:
The antioxidant capacity of naringenin shows how it can interact with some proteins, either by reduction ROS, chelation of metals or in a pro-oxidant manner, which in a cell metabolism altered by cancer would promote apoptosis and eventually inhibition of cell proliferation. In addition to these mechanisms, naringenin has shown several others by which it inhibits proliferation. These include some signaling pathways such as EGFR/MAPK, Akt, and the Wnt/β-catenin pathway, among others. [30,31] (Page 14, lines 320-325).
- Kachadourian, R.; Day, B.J. Flavonoid-induced glutathione depletion: potential implications for cancer treatment. Free Radic Biol Med. 2006 1;41(1):65-76.
- Ghanbari-Movahed, M.; Jackson, G.; Farzaei, M.H.; Bishayee, A. A. Systematic Review of the Preventive and Therapeutic Effects of Naringin Against Human Malignancies. Front Pharmacol. 2021 29;12:639840.

Round 2
Reviewer 2 Report
The authors provided a revised version of their work. I believe that the manuscript is suitable for publication in the present form.